# Microwave-Assisted Optimization of Polyvinyl Alcohol Cryogel (PVA-C) Manufacturing for MRI Phantom Production

**DOI:** 10.3390/bioengineering12020171

**Published:** 2025-02-10

**Authors:** Ivan Vogt, Martin Volk, Emma-Luise Kulzer, Janis Seibt, Maciej Pech, Georg Rose, Oliver S. Grosser

**Affiliations:** 1Research Campus STIMULATE, Otto-von-Guericke-University Magdeburg, 39106 Magdeburg, Germany; 2Faculty of Electrical Engineering and Information Technology, Otto-von-Guericke-University Magdeburg, 39106 Magdeburg, Germany; 3Department of Radiology and Nuclear Medicine, University Hospital Magdeburg, 39120 Magdeburg, Germany; 4Department Biomedical Magnetic Resonance, Faculty of Natural Sciences, Otto-von-Guericke-University Magdeburg, 39106 Magdeburg, Germany

**Keywords:** microwave-based manufacturing, phantom material, polyvinyl alcohol cryogel, MRI

## Abstract

Objectives: Anthropomorphic phantoms offer a promising solution to minimize animal testing, enable medical training, and support the efficient development of medical devices. The adjustable mechanical, biochemical, and imaging properties of the polyvinyl alcohol cryogel (PVA-C) make it an appropriate phantom material for mimicking soft tissues. Conventional manufacturing (CM) of aqueous solutions requires constant stirring, using a heated water bath, and monitoring. Methods: To explore potential improvements in the dissolution of PVA crystals in water, a microwave-based manufacturing method (MWM) was employed. Samples created using CM and MWM (*n* = 14 each) were compared. Because PVA-C is a multifunctional phantom material (e.g., in magnetic resonance imaging (MRI)), its MRI properties (T1/T2 relaxation times) and elasticity were determined. Results: T1 relaxation times did not significantly differ between the two methods (*p* = 0.3577), whereas T2 and elasticity for the MWM were significantly higher than those for the CM (*p* < 0.001). The MWM reduced the production time by 11% and decreased active user involvement by 93%. Conclusions: The MWM offers a promising, easily implementable, and time-efficient method for manufacturing PVA-C-based phantoms. Nevertheless, manufacturing-related microstructural properties and sample molding require further study.

## 1. Introduction

Testing the functionality and safety of medical devices as part of an approval procedure often involves in vivo studies. In addition to ethical concerns, the high cost and complexity of preclinical and clinical trials must also be considered [1]. Hence, endorsement of substitutes for in vivo experimentation, as demonstrated by initiatives such as the Center for Alternatives to Animal Testing, is required [2]. Anthropomorphic phantoms customized for specific purposes offer versatile solutions to these challenges by enabling the development of diverse medical devices and facilitating efficient medical training [3]. These phantoms are particularly useful in areas such as image-guided minimally invasive procedures, where the precise simulation of human anatomy and realistic interactions with medical instruments are crucial. In this context, the adjustable mechanical, biochemical, and imaging properties of a nontoxic and cost-effective polyvinyl alcohol cryogel (PVA-C) have been demonstrated for use as a phantom material for the replication of soft tissue based exclusively on frozen, water-soluble PVA fibers [4,5].

Conventional manufacturing (CM) of PVA-water solutions is typically performed using a heated water bath combined with a stirrer [6,7]. This method relies on conductive heating. Thermal effects cause a gradual dissolution in water, leading to an increase in viscosity. This process may result in the agglomeration and adherence of PVA to the vessel and stirrer as a consequence of accelerated or uneven heating, or inadequate stirring. This may result in the accumulation of undissolved PVA [7,8]. Molecular dynamics simulations [9] have demonstrated that PVA chains exhibit a propensity to form aggregates in aqueous solutions, a phenomenon driven by hydrophobic effects and intramolecular hydrogen bonding. The CM uses continuous stirring to prevent lump formation and ensure complete dissolution. However, stirring requires constant monitoring and can introduce air bubbles or cause localized agglomeration. Mechanical action can also cause lumps to adhere to the equipment, thereby altering the local PVA/water ratio. Undispersed lumps can persist through subsequent processing steps, potentially affecting production results. This highlights the delicate balance required in the CM to achieve optimized PVA dissolution.

Alternatively, a microwave-based method (MWM) offers several advantages over CM. As described by Kappe [10], MWM uses “in-core volumetric heating”, where microwave energy is directly coupled to the molecules in the mixture. Further simulations and measurements showed that this microwave approach provides rapid heating throughout the solution, in contrast to conventional methods [11]. As a result, this method eliminates the need for constant manual stirring. The absence of mechanical stirring also minimized the risk of introducing air or transferring undissolved particles.

The aim of this study was to validate samples manufactured using the MWM for use in MRI. The analysis was performed by analyzing typical parameters describing material properties in MRI (e.g., T1 and T2 relaxation), as well as elasticity measurements to assess the mechanical behavior relevant to haptic properties. A set of samples manufactured after CM was used as a control to identify the effects of MWM in comparison with the actual standard process.

## 2. Materials and Methods

### 2.1. Preparation of Samples

For each manufacturing process, 14 batches of 200 mL, each with a PVA mass content of 8 wt.%, were prepared (Kuraray Poval^®^ PVA 15–99, Kuraray Europe GmbH, Hattersheim am Main, Germany). Two PVA-C samples were extracted from each batch using cylindrical silicone molds (Ø 40 mm × 15 mm). A standard hotplate (Rotilabo MH 20; Carl Roth GmbH + Co., KG, Karlsruhe, Germany) was used for CM. The PVA aqueous solution was dissolved in a beaker, which was placed in a temperature-controlled water bath maintained at 95 °C. Manual stirring was conducted using a laboratory spoon. For MWM, a microwave oven (Comfee CMG20XS, Midea Europe GmbH, Eschborn, Germany) with an output of 240 W was employed. To control water evaporation in MWM, a microwaveable lid with a small vapor outlet hole was used. Due to the use of a stirrer in CM, this arrangement was not possible. For both methodologies, following the dissolution of the PVA crystals, the evaporated water was replaced with water at a temperature of 95 °C to maintain the desired mass ratio. The PVA solution was briefly mechanically stirred to ensure uniform mixing. Subsequently, the mixture was allowed to cool to room temperature (20 °C), which facilitated the release of the trapped air bubbles.

To satisfy the methodological requirements for evaluating elasticity, the molds were covered with 3D-printed stamps, thus ensuring the reproducibility of the sample heights and planar surfaces. As stated by Peppas and Stauffer, PVA in an aqueous medium is crosslinked by freeze–thaw cycles, resulting in the formation of a 3-dimensional structure [12,13]. To create a cryogel in this study, a single freeze–thaw cycle was conducted, comprising 12 h at −18 °C and 12 h at 21 °C. The resulting material is characterized by its translucent, milky appearance (Figure 1A).

### 2.2. Estimation of T1 and T2

MRI was conducted using a 3 T MRI system (Siemens MAGNETOM Skyra, Siemens Healthineers AG, Forchheim, Germany) with a magnetization-prepared rapid gradient echo imaging sequence to determine the T1 relaxation time, employing inversion times ranging from 516 to 6500 ms (see Figure 1B). A spin-echo sequence with echo times of 12–1000 ms was employed for T2 analysis (see Figure 1C). The acquisition parameters included a field of view of 250 mm × 250 mm, matrix size of 128 × 128, and slice thickness of 6 mm for T1 and 4.5 mm for T2.

To determine T1 and T2, four regions of interest (ROI), each comprising 7 × 7 pixel values, were employed for exponential interpolation of the corresponding relaxation times [14]. The script then extracted the ROIs and performed an exponential fit of the mean intensity values dependent on the TE or TI. The base functions for the fits were defined using the theoretically expected curve and a confidence interval (CI) of 95%. The underlying absolute value function for T1 evaluation was as follows:(1)$$f(x)=a\cdot \left|1-2\cdot \exp\left(-\frac{x}{T1}\right)\right|$$

To achieve a more optimal regression convergence, the initial parameters were set to *a* = 100 and T1 = 2000 ms within the algorithm. T2 was fitted using a simple exponential function. Any discernible heterogeneity resulting from local air bubbles was excluded from the ROI analysis.

### 2.3. Estimation of Young’s Modulus, E

The samples were subjected to mechanical compression at a speed of 10 mm/min using a compression-testing machine (Xforce HP 50 N, ZwickiLine Z0.5TN; ZwickRoell GmbH, Ulm, Germany). The Young’s modulus, *E*, was determined using the secant modulus at 5% and 10%. Each sample was compressed four times to analyze its respective *E*. In contrast to ROI analysis, the entire sample size was measured.

Consequently, some samples were removed (e.g., based on visible local air bubbles on the surface; see Table 1), resulting in 17 samples (*n* = 68 data points) for the CM and 24 samples (*n* = 96) for the MWM.

### 2.4. Statistics

MATLAB (MathWorks Inc., Natick, Massachusetts, U.S., version 24.1.0 [R2024a] with Statistics and Machine Learning Toolbox) was used for the statistical analysis. Statistical analysis of the significance of T1 and T2 refers to all four ROIs for each sample from a batch (each *n* = 112). Given the non-normal distribution of the determined parameters (Kolmogorov–Smirnov test, *p* < 0.001), a nonparametric test (two-sided Wilcoxon rank-sum test) was employed to identify significant effects resulting from the manufacturing process on T1, T2, and *E*. To determine the production time, the duration and active user contribution of the main process (from the unsolved solution to the final mixture) were measured for the last batch after the user became familiar with the manufacturing processes. All tests were two-sided, and significance was set at *p* < 0.05.

## 3. Results

The data summarized in Table 1 form the basis for conducting statistical analyses across all batches and resulting samples. The individual data points for all samples used in the statistical tests are available for download in the Appendix A.

This study employed a comparative approach, examining both methods across T1, T2, and *E* (see Figure 2). 

Additionally, the absolute differences in the medians (ADMs) and relative absolute deviation of the medians (RDM) were calculated for each case. Significant differences were observed in T2 (ADM = 22; RDM = 13.62%) and *E* values (ADM = 0.14; RDM = 5.84%, both *p* < 0.001). However, the T1 results (*p* = 0.3577) showed no statistically significant differences between the manufacturing methods, with only minor variations (ADM = 4; RDM = 0.28%). The standard deviations of all evaluated parameters were lower with MWM than with CM. For the interpolated relaxation times with mean systematic errors of all corresponding ROIs for each sample, the following CIs were calculated: MWM with T1_CI,MWM_ = ±63.5 ms and T2_CI,MWM_ = ±5.3 ms; and CM with T1_CI,CM_ = ±60.9 ms and T2_CI,CM_ = ±4.3 ms.

Regarding the production process, the MWM reduced the total manufacturing time by 5 min (CM: 45 min; MWM: 40 min), which represents an 11% reduction. More importantly, the MWM drastically decreased the active user involvement time by 93% (CM: 45 min; MWM: 3 min of active manual stirring). The water loss for the MWM is approximately 4%, while for the CM, it is 10%.

## 4. Discussion

This study examines an alternative manufacturing approach for producing PVA-C-based samples for MRI phantoms. The impact of the manufacturing process on typical MRI parameters such as T1 and T2 was analyzed. The Young’s modulus was tested as a surrogate for the tactile properties to assess the suitability of the phantoms for use in interventional procedures. This study employed a setting with multiple production batches to evaluate the reproducibility of the results.

The results showed no statistically significant differences in T1 between the two methods. However, the T2 and elasticity values were significantly higher for MWM than for CM. These differences could indicate variations in the microstructure of the resulting PVA-C with potentially different water distributions, smaller pore characteristics, and higher crosslinking densities compared to those of the CM. These microscale differences may explain the observed variations in MRI properties and mechanical behavior. Furthermore, the standard deviations of all evaluated parameters were lower for the MWM than for the CM, suggesting higher homogeneity and manufacturing reproducibility.

The MWM for PVA dissolution and subsequent cryogel production offers several advantages over CM. The use of the MWM has been shown to reduce manufacturing time and require less active steering, resulting in less user involvement. In addition, MWM simplifies the process by using a basic microwave, eliminating the need for specialized equipment and making this method more accessible and cost-effective.

The volumetric heating provided by the MWM allowed for more uniform dissolution of the PVA without the need for manual stirring, reducing the number of air bubbles trapped within the solution. This improvement is primarily attributed to the absence of constant stirring, which typically incorporates more air into the solution during CM processes. Furthermore, the transfer of the prepared PVA solution to individual sample molds can also result in the introduction of air bubbles, which may occur when using both the MWM and CM techniques. This necessitated additional refinement to ensure consistent phantom quality. The presence of air bubbles introduced random measurement errors in both methods. To mitigate their impact, the samples with visible air entrapment were excluded from the elasticity analysis. In contrast, these bubbles did not affect the T1 and T2 measurements because they were not included in the ROIs for the interpolation process. To address the issue of air bubbles, future studies could explore degassing methods such as centrifugation [6] of the solution or ultrasonication [15] after molding the PVA liquid.

Furthermore, both methods result in water loss during heating. The MWM minimized water loss using a microwave-suitable cover with a small vapor outlet hole, in comparison to the CM. For CM, this was attributed to the fact that it was not possible to cover the liquid continuously when using a manual stirrer. As a result, more water had to be added, which increased the production effort. Initially, temperature monitoring within the microwave was not necessary because of the sealed environment, which limited water loss compared to the open manual stirring setup in CM.

While the MWM offers several advantages in terms of PVA dissolution, acknowledging the limitations of this approach and identifying areas that require further investigation are crucial. Zhao et al. [11] provided insights into the uneven temperature distribution that can occur when liquids are heated in a microwave oven, demonstrating the phenomenon of temperature stratification with the formation of hotter top layers. This is consistent with the mathematical overview of Hill and Marchants [16] which indicated that hotspots can result from temperature-dependent material properties and nonuniform heating. Nevertheless, the possibility of hotspot formation and microscopic structural changes resulting from localized heating persists despite the absence of macroscopic visible aggregations observed in the present study. The impact of microwave radiation on the PVA structure has been investigated in several contexts. A previous study on PVA in ethylene glycol demonstrated that only minor changes were observed after 60 min of microwave irradiation at 700 W [17]. Subsequent studies documented the structural modifications of solid PVA films as a result of microwave exposure [18,19]. This emphasizes the complexity of microwave–material interactions and the necessity for further investigation of the specific effects on PVA in aqueous solutions to optimize the method and gain insights into its effects on the final phantom product. Thermal-imaging cameras can be employed to analyze the temperature distribution throughout the PVA solution during and after heating, thereby revealing potential hotspots or uneven heating patterns that may arise.

Further investigation is required to elucidate the relationship between microwave power and heating efficiency. Higher energy levels are known to facilitate the boiling of water and enhance the dissolution rates of PVA crystals, which can shorten the production time.

The effect of MWM on cryogel formation, particularly in terms of pore size and distribution, remains unclear. To address this issue, future research could employ techniques [20] such as scanning electron microscopy to analyze the impact of different manufacturing methods on the properties of PVA-based cryogel. Additionally, Fourier-transform infrared spectroscopy could be used to investigate potential changes in chemical structure, while thermogravimetric analysis could provide insights into the thermal stability. This methodology could also provide relevant information on the degree of crosslinking of cryogels using different production methods. It might also provide data about water content and decomposition temperature, which could be relevant for understanding the resulting cryogel structure. These techniques would offer a comprehensive characterization of the overall properties of the cryogels, allowing for a more thorough understanding of the effects of MWM on PVA-C formation. Furthermore, a comprehensive approach is required to investigate not only microwave-related factors such as microwave type, geometry, and vessel characteristics, but also the intrinsic properties of PVA itself. This should include an examination of how PVA properties, such as molecular weight, concentration, and degree of hydrolysis, interact with microwave heating to affect manufacturing and cryogel outcomes. Such a detailed investigation could provide a deeper understanding of the complex interaction between the material properties and processing conditions in microwave-assisted production.

Finally, we have to consider some application-specific aspects in further tests. The suitability of the manufactured material is also determined by the haptic impression. We suggest that application-specific tests are performed to determine whether the observed differences have a significant impact on the intended uses of the material (e.g., by experienced interventionalists who routinely perform tissue punctures).

## 5. Conclusions

This study presents an alternative microwave-based PVA-C manufacturing method that demonstrates reproducibility comparable to that of conventional methods with regard to MRI properties and mechanical elasticity. However, the process is considerably more time-efficient, eliminating the requirement for specialized production equipment. Nevertheless, further studies are required to elucidate the manufacturing-related microscopic effects of microwave irradiation on the PVA structure and to optimize the heating parameters. The MWM enables a standardized method to enhance innovation and broaden the application of PVA-based phantoms for research, medical training, and small-scale device development.

## Figures and Tables

**Figure 1 bioengineering-12-00171-f001:**
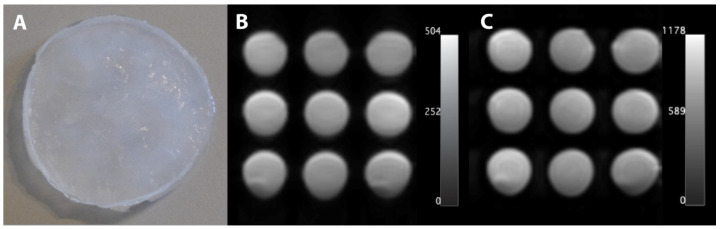
(**A**) Photo of a PVA-C sample, (**B**) T1-weighted image (T1 relaxation time = 3000 ms), and (**C**) spin echo for T2-weighted image (T2 relaxation time = 250 ms).

**Figure 2 bioengineering-12-00171-f002:**
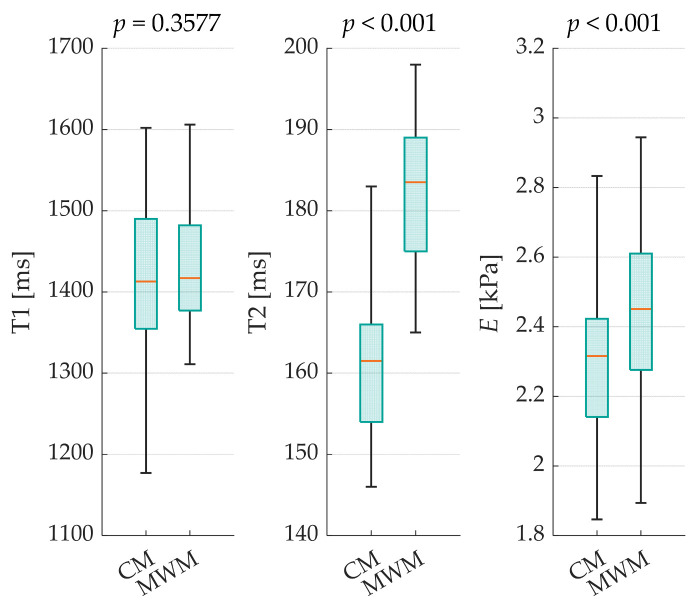
Effect by manufacturing methods on (**left**) T1, (**middle**) T2, and (**right**) *E; Boxplot* with orange line as median.

**Table 1 bioengineering-12-00171-t001:** T1, T2, and *E* values for each manufacturing batch.

Batch No.	CM					MWM				
	T1 (ms)	T1_CI_ (ms)	T2 (ms)	T2_CI_ (ms)	*E* (kPa)	T1 (ms)	T1_CI_ (ms)	T2 (ms)	T2_CI_ (ms)	*E* (kPa)
1	1309.9	43.6	158.9	3.8	2.54	1416.6	85.4	190.4	5.5	2.38 *
2	1328.4	45.4	172.1	4.1	2.38	1455.9	79.5	187.9	5.5	2.83 *
3	1244	48.5	152.4	4.3	-	1489.6	71.6	191.8	6.5	-
4	1318.3	40.8	158	3.5	2.62	1487.8	69.8	192.4	5.5	2.6
5	1522.9	71.9	183.9	6.9	-	1455.9	81.5	184.5	5.1	2.22
6	1477.5	75.3	160.5	5.3	2.45 *	1499.3	73.6	189.3	6.6	2.17
7	1490.9	69	164.8	6.4	2.26	1384.9	56.9	179.1	3.9	2.62
8	1466	75	168.3	4.8	2.13 *	1394.8	50	175.1	5.4	2.51
9	1455.6	79.1	164.3	4.3	-	1386.6	55.1	178.8	3.5	2.39
10	1492.6	71	154.1	4.1	2.18 *	1378.3	56.9	176.5	3.8	2.61
11	1402.4	57.6	159.8	2.8	2.31	1388.9	51.9	175	5.4	2.46
12	1435.1	57.5	149.4	2.8	2.15	1423.9	50.9	178.1	6	2.35
13	1385	58.1	153.6	3.4	2.1	1435.6	53.9	176.4	5.4	2.61
14	1466	61	165.3	3.8	-	1434.9	51.6	178.4	6	2.28

T1 and T2 represent the means of two samples, each comprising four ROIs; *E* represents four compressions. T1_CI_ and T2_CI_ describe the uncertainty of the relaxation time interpolation within a 95% confidence interval (CI). Because of visible inhomogeneities and air bubbles introduced during production, only one sample per batch was included in the analysis (*); in some cases, no sample was included (-).

## Data Availability

The processed data supporting the conclusions of this study are available in the Appendix A of the article. The raw data (MRI images) will be made available by the authors upon request. Justification for the limited availability of raw data: The processed data in the Appendix A are sufficient to understand and verify the results and conclusions of our study.

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
