# Peer review of "Microwave-Assisted Optimization of Polyvinyl Alcohol Cryogel (PVA-C) Manufacturing for MRI Phantom Production"

_bioengineering, 2025, doi:10.3390/bioengineering12020171_

Round 1
Reviewer 1 Report
Comments and Suggestions for Authors
The study titled as “Microwave-Assisted Optimization of Polyvinyl Alcohol Cryo-2 gel (PVA-C) Manufacturing for MRI Phantom Production” is an interesting study, and has promising results for MRI studies. However, the study still seems very superficial, because the characterization part is very limited. Following major revisions, it may be published in the journal Bioengineering.
- First of all, the details about synthesis process for both conventional (CM) and microwave-assisted (MWM) ones have to be given. The stirring time for CM, the amount of evaporated water. Also, the duration of exposure in the microwave, how carbonization is prevented in the microwave, and how water evaporation is prevented should be explained for MWM. Also the number of freeze-thawing process should be given. If it is one, the effect of number of freeze-thawing process should be studied.
- I don't understand what the values ​​given in the supplement file mean, but they do not match the values ​​given in the article.
- The FT-IR, TGA, SEM images, and most important gel permeation chromatography for prepared PVA cryogels synthesized via CM and MWM should be compared.
- The reason of observing higher T2 values for MWM than CM should be explained.
- The related study https://doi.org/10.3390/polym14010070 should be cited.
Author Response
We sincerely appreciate your thorough review and valuable feedback on our manuscript Microwave-assisted optimization of polyvinyl alcohol cryogel (PVA-C) manufacturing for MRI phantom production. We are pleased that you find our study interesting and recognize its potential for MRI studies. We acknowledge your concerns regarding the depth of characterization in our current manuscript.
In the following answers, we will address your points comprehensively and outline the steps we have taken to enhance the quality of the manuscript. Please note that the lines in the answers already refer to the improved manuscript.
REVIEWER 1:
First of all, the details about synthesis process for both conventional (CM) and microwave-assisted (MWM) ones have to be given. The stirring time for CM, the amount of evaporated water. Also, the duration of exposure in the microwave, how carbonization is prevented in the microwave, and how water evaporation is prevented should be explained for MWM. Also the number of freeze-thawing process should be given. If it is one, the effect of number of freeze-thawing process should be studied.
Answer: Thank you for your insightful feedback. We provide additional facts on the synthesis processes for both conventional and microwave-assisted methods.
For the CM, stirring was continuous throughout the entire 45-minute process, as stated in the manuscript: "Regarding the production process, the MWM time was reduced by 11% (CM: 45 min; MWM: 40 min) and required 93% less active stirring (CM: 45 min; MWM: 3 min) (line 158-159)". For the microwave-assisted method MWM, we utilized a microwave lid/cover with a small vapor outlet hole to minimize water evaporation. This was not feasible for CM due to the manual stirring (line 84-86).
The amount of evaporated water was approximately 4% for MWM and 10% for CM. We did not observe any signs of carbonization during the MWM process. Regarding the freeze-thaw process, we conducted a single cycle (one time freeze, one time thaw), as mentioned in the manuscript (line 95-96).
However, we recognize that further optimization of both methods has potential. Future studies could explore the effects of multiple freeze-thaw cycles on the properties of the resulting PVA-C. It is important to note that the number and conditions of freeze-thaw cycles can significantly influence the resulting cryogels [13]. Future research could investigate the impact of varying freeze-thaw cycles on cryogels produced by e.g. MWM. However, the focus was on an initial comparison of the production methods.
Question: I don't understand what the values ​​given in the supplement file mean, but they do not match the values ​​given in the article.
Answer: We apologize for any confusion caused by the apparent discrepancy between the values in the main article and the supplementary file. In general: the data presented in the main paper are aggregated per batch (for each CM and MWM), providing an immediately recognizable overview of the results. The supplementary file contains the raw data for all individual measurements.
We decided on this approach to avoid an overly complex and unclear table in the main manuscript, as some columns would have contained over 100 values each. In our opinion, this separation allows for a clearer presentation in our main article while providing full transparency and access to all data in the supplementary materials file for those who wish to examine the results in detail. We have updated our supplementary files to make the data contained therein easier to interpret and thank you for bringing this to our attention and hope that this statement clarifies the apparent inconsistencies.
Question: The FT-IR, TGA, SEM images, and most important gel permeation chromatography for prepared PVA cryogels synthesized via CM and MWM should be compared.
Answer: We appreciate your suggestion to include additional characterization methods such as FT-IR, TGA, SEM imaging, and gel permeation chromatography for comparing the resulting PVA cryogels synthesized via CM and MWM. While these techniques would without doubt provide valuable insights into the e.g. microstructure and chemical properties of the cryogels, we decided to focus our study on the MRI properties and mechanic. These are directly relevant to the application of this material as a phantom material. Including the suggested additional analyses would have significantly expanded the scope of the study beyond its original intent as a brief report format. Moreover, a comprehensive investigation incorporating these techniques, along with an exploration of e.g. different microwave power settings to further optimize the process, would constitute a separate, more extensive study. We acknowledge the importance of these additional characterization methods and have already noted in our discussion that future research should include techniques such as SEM (line 233) to further elucidate the effects of different manufacturing methods on PVA-C properties. This approach allows us to maintain the focus of the current study while setting the stage for more detailed investigations in future work.
We have also included the other techniques you mentioned in our discussion and thank you for your valuable input. We appreciate your contribution, which has helped to improve the quality and scope of our research discussion.
The reason of observing higher T2 values for MWM than CM should be explained.
Answer: The explanation for the higher T2 values observed in the MWM compared to the CM is briefly given in the discussion section (lines 175-178). We hypothesize that these differences could be attributed to variations in the microstructure of the resulting PVA-based cryogels. This could include different water distributions, smaller pore characteristics, and higher cross-linking densities. We recognise that a more detailed investigation of these microstructural differences would be valuable. Although beyond the scope of this brief report, we fully agree that additional techniques, such as the mentioned FT-IR, TGA, SEM images, and gel permeation chromatography, could provide deeper insights into the structural changes induced by the different manufacturing methods, especially with respect to the microwave heating. This may be an interesting direction for future research to elucidate the exact mechanisms underlying the observed differences in T2 values between MWM and CM.
The related study https://doi.org/10.3390/polym14010070 should be cited.
Answer: We appreciate your suggestion to include the reference to the study https://doi.org/10.3390/polym14010070. We will certainly incorporate this citation in our revised manuscript, as it provides valuable context and insights relevant to our work.
Reviewer 2 Report
Comments and Suggestions for Authors
1. For better reproducibility, the authors should measure the stability of samples created using CM and MWM.
2. According to sentenses 169-170 ( The results showed no statistically significant differences in T1 between the two methods. However, the T2 and elasticity values were significantly higher for MWM than for CM.), what is the impact on applications?
3. Is it possible to use mechanical stirring instead of manual stirring? If so, then the description of sentences 178-179 are not appropriate.
4. The MWM actually reduced the production time by 5 min. Sentences 240-241 (this process is quite time saving) do not describe properly.
5. Some small mistakes found in the manuscript, e.g., non-uniform reference format, ref. 8, 11, 14, 15, 16, 18, 19.
Author Response
We sincerely appreciate your thorough review and valuable feedback on our manuscript Microwave-assisted optimization of polyvinyl alcohol cryogel (PVA-C) manufacturing for MRI phantom production. We are pleased that you find our study interesting and recognize its potential for MRI studies. We acknowledge your concerns regarding the depth of characterization in our current manuscript.
In the following answers, we will address your points comprehensively and outline the steps we have taken to enhance the quality of the manuscript.
For better reproducibility, the authors should measure the stability of samples created using CM and MWM.
Answer: Thank you for your insightful comment regarding the stability of samples created using CM and MWM. We are grateful for the opportunity to clarify and expand this aspect of our research in this study. While we did measure the Young's modulus for several batches and the resulting probes produced by both methods, we acknowledge that further investigation into e.g. long-term stability would indeed enhance the reproducibility and applicability of our findings. In our observations, we did not detect any perceptible visual or tactile differences between samples produced by CM and MWM.
However, we agree that a more comprehensive analysis of temporal and mechanical stability would be valuable. Future research could involve e.g. measuring stability at various time points (e.g., hours, days, and weeks after production) to assess any potential changes in the PVA-C over time (e.g., through water loss). In addition, mechanical stability tests, especially those that simulate real-life applications such as needle insertion for training scenarios, would provide important insights into the practical effects of any differences between those samples. While we found a measurable technical difference between CM and MWM in terms of T2 relaxation times and elasticity, the practical impact of these differences on material interaction and performance in specific applications remains to be fully clarified. We believe that further tests with needles and other relevant instruments would be a good next step to bridge this gap in understanding. These additional studies would not only address stability concerns. They would also provide a more comprehensive comparison of the two methods. They may reveal differences that are not immediately apparent from visual or tactile examination.
We appreciate your suggestion and agree that these additional studies would significantly contribute to the reproducibility and practical applicability of our findings.
According to sentenses 169-170 (The results showed no statistically significant differences in T1 between the two methods. However, the T2 and elasticity values were significantly higher for MWM than for CM.), what is the impact on applications?
Answer: Thank you for your follow-up question regarding the impact of the observed differences in T2 and elasticity values between CM and MWM on potential applications. Based on our previous response, we agree that the statistically significant differences in T2 relaxation times and elasticity values between CM and MWM samples may have an effect on particular applications. However, the practical significance of these differences needs to be carefully considered in the context of the specific use cases. For MRI applications, the higher T2 values observed in MWM samples might affect the contrast in T2-weighted imaging which could lead to small variations in image appearance or quantitative measurements when using phantoms produced by CM/MWM. The absence of a significant difference in T1 values suggests minimal impact on T1-weighted imaging in the MWM samples. In terms of elasticity, the higher values observed in the MWM samples could influence the haptic feedback in e.g. training scenarios or the mechanical response in certain test applications (e.g., needle interaction). These differences could be particularly relevant for applications where precise tissue mimicry is required. It depends on the specific requirements of the use-case.
To address this, we suggest that application-specific testing be performed to determine whether the observed differences significantly affect different intended uses of the material (e.g. by experienced interventionalists who routinely perform tissue punctures).
Is it possible to use mechanical stirring instead of manual stirring? If so, then the description of sentences 178-179 are not appropriate.
Answer: Thank you for your question regarding the use of mechanical stirring instead of manual stirring in the CM procedure. While mechanical stirring is indeed an alternative, we have chosen to use manual stirring for a number of different reasons. Firstly, greater control over the mixing process can be achieved by manual stirring. This is important when dissolving the PVA crystals. This method allows the operator to closely monitor the consistency and increasing viscosity of the solution and adjust the stirring speed and technique in real time. The aim is to ensure the most uniform dissolution possible and reduce the risk of agglomeration or undissolved particles. In addition, mechanical stirring can introduce air bubbles into the solution, which can affect the quality of the final product. This is because if the stirring speed is too high, it can lead to excessive air entrainment, which further complicates the material properties that we want to optimize. Achieving a balance between effective mixing and minimising air entrapment is therefore crucial. In addition, one of our main objectives was to produce phantoms with a minimum of additional equipment.
The MWM actually reduced the production time by 5 min. Sentences 240-241 (this process is quite time saving) do not describe properly.
Answer: Thank you for pointing out this discrepancy. You are correct. The current phrasing does not accurately reflect the time savings achieved by the MWM. We have therefore revised the text to describe the time reduction in more detail.
Some small mistakes found in the manuscript, e.g., non-uniform reference format, ref. 8, 11, 14, 15, 16, 18, 19.
Answer: Thank you for your careful review and for pointing out the inconsistencies in our references section. We appreciate your attention to detail. We have carefully reviewed and revised all references in the manuscript to ensure a consistent citation style. We have made the necessary corrections to bring them into line with our chosen citation format.
Round 2
Reviewer 1 Report
Comments and Suggestions for Authors
Thank you for your great effort to complete required revisions.